# Antagonization of Ghrelin Suppresses Muscle Protein Deposition by Altering Gut Microbiota and Serum Amino Acid Composition in a Pig Model

**DOI:** 10.3390/biology11060840

**Published:** 2022-05-30

**Authors:** Xiaoxi Yan, He Zhang, Ailian Lin, Yong Su

**Affiliations:** 1Jiangsu Key Laboratory of Gastrointestinal Nutrition and Animal Health, College of Animal Science and Technology, Nanjing Agricultural University, Nanjing 210095, China; 2020105060@stu.njau.edu.cn (X.Y.); 2018205020@njau.edu.cn (H.Z.); 2020105059@stu.njau.edu.cn (A.L.); 2National Center for International Research on Animal Gut Nutrition, Nanjing Agricultural University, Nanjing 210095, China

**Keywords:** ghrelin, feeding behavior, gut microbiota, serum amino acids, muscle protein deposition

## Abstract

**Simple Summary:**

This study investigated the effects of the antagonization of ghrelin on muscle protein deposition, eating patterns and gut microbiota in pigs by injecting ghrelin antagonist ([D-Lys3]-GHRP-6) in a short term. We found that the antagonization of ghrelin affected the eating patterns of animals, which resulted in changes in the absorption of amino acids and gut microbiota, and it reduced protein deposition in muscles. We emphasize the important role of ghrelin in promoting muscle protein deposition and provide new clues for future research on improving muscle loss.

**Abstract:**

Ghrelin is an appetite-stimulating hormone that can increase food intake and has been reported to prevent muscle loss; however, the mechanism is not yet fully understood. In this study, [D-Lys3]-GHRP-6 (GHRP) was used to investigate the effects of the antagonization of ghrelin on muscle protein deposition, eating patterns and gut microbiota in a pig model. We found that the growth performance and muscle fiber cross-sectional area of pigs treated with GHRP were significantly reduced compared with the control (CON) group. Moreover, the levels of serum isoleucine, methionine, arginine and tyrosine in the GHRP group were lower than that of the CON group. The abundance of acetate-producing bacteria (*Oscillospiraceae* UCG-005, *Parabacteroides* and *Oscillospiraceae* NK4A214 group) and acetate concentration in the colons of pigs treated with GHRP were significantly reduced. In addition, the injection of GHRP down-regulated the mRNA expression of *MCT-1* and *mTOR*, and it up-regulated the mRNA expression of *HDAC1*, *FOXO1* and *Beclin-1*. In summary, the antagonization of ghrelin reduced the concentration of important signal molecules (Arg, Met and Ile) that activate the mTOR pathway, concurrently reduce the concentration of HDAC inhibitors (acetate), promote autophagy and finally reduce protein deposition in muscles.

## 1. Introduction

The muscles of agricultural animals are the main sources of meat products, providing humans with high-quality animal protein and nutrients [1]. Protein deposition in muscles is regulated by diverse regulatory factors, such as amino acids, myogenic regulatory factors (MyoD, MyoG, etc.) and muscle degradation factors (MuRF-1 and Atrogin-1) [2,3]. Recent studies have found that ghrelin plays an important role in muscle growth and protein deposition [4,5]. Ghrelin is a peptide hormone composed of 28 amino acids, which increases the food intake of animals [6,7]. The level of circulating ghrelin can reflect the energy state, which increases during fasting and decreases after feeding [8]. The injection of ghrelin activates insulin receptor substrate 1 (IRS-1), as well as its downstream signal molecule phosphatidylinositol-3-kinase (PI3K), and the target of rapamycin (mTOR), which play important roles in protein synthesis [9,10]. The mTOR regulates the phosphorylation of downstream effectors ribosomal S6 kinase 1 (S6K1) and ribosomal protein S6 (rpS6) and promotes protein synthesis in pigs [9,11]. In addition, studies have also found that ghrelin treatment inhibits the catabolism of C2C12 myotubes mediated by inflammatory factors, promotes its differentiation and fusion and increases protein deposition in muscles [12,13]. However, ghrelin treatment directly promotes the expression of myogenic regulatory factors (MyoD, MyoG), reduces the expression of protein degradation factors (Atrogin-1, MuRF1) and promotes the expression of mitochondrial functional genes, such as differentiation and fusion, which promote protein deposition in mouse muscles [3]. Interestingly, this study also found that ghrelin has a significant effect on gut microbial composition. Compared with WT mice, the relative abundance of butyrate-producing bacteria *Roseburia* and *Clostridium* XIVb in Ghrl^−/−^ mice is significantly reduced, and butyrate plays an important role in reducing inflammation and muscle loss. Previous studies mainly focused on the direct injection of ghrelin or the construction of the gene knockout model to explore the effect of ghrelin on protein deposition in muscles. Therefore, this study chose to reverse prove the effect of ghrelin on muscles by injecting ghrelin antagonist ([D-Lys3]-GHRP-6), which has been used to explore the effect of ghrelin in other studies.

In recent years, the interaction between the gut and muscles has gradually become a hot spot in the research direction of human and animal health. The “gut–muscle axis” has been innovatively proposed, i.e., gut microbiota and their metabolites affect muscle function and metabolism to a certain extent [14]. Short-chain fatty acids (SCFAs) are the principal end products of colonic microbiota and are widely considered to mediate the interaction between gut microbiota and muscles [15]. One study found that, after transplanting pig fecal microbiota into germ-free mice, the mice showed skeletal muscle fiber characteristics consistent with those of donors [16]. In addition, SCFAs cocktail can reduce dexamethasone-induced muscle atrophy in mice [17]. Both acetate and butyrate have been shown to inhibit histonedeacetylases (HDACs) in various tissues or cells [18,19]. Among them, HDAC1 has been shown to promote skeletal muscle atrophy in response to nutritional deficiency [20]. This indicates that micronutrients and metabolites derived from gut microbiota can reach and act on muscles [21].

In this study, we linked ghrelin, gut microbiota and protein deposition in muscles. We proposed the hypothesis that the antagonization of ghrelin affects the eating patterns of animals, which results in changes in the absorption of amino acids and gut microbiota and affects protein deposition in muscles. Therefore, the purpose of this study was to investigate the effects of the antagonization of ghrelin on the eating patterns, gut microbiota and muscle protein deposition in a pig model.

## 2. Materials and Methods

### 2.1. Animals and Experimental Design

The trial was carried out under the supervision of the Animal Care and Use Committee of the Nanjing Agricultural University in Nanjing, Jiangsu, China [ethic code: SYXK (SU) 2017–0007]. A total of 12 52-day-old barrows (Duroc × Landrace × Large White) with similar body weights (initial body weight: 17.53 ± 0.41 kg) were randomly divided into the control group (CON) and the ghrelin antagonist ([D-Lys3]-GHRP-6) group (GHRP). Each group consisted of 6 replicates (pens) with one pig per pen. The pigs were maintained in a 12:12 h light (7:00–19:00)–dark cycle. The daily feeding amount of each pig was consistent (90% of ad libitum), and they were fed at seven in the morning. Water was provided ad libitum over the 7 d experimental period. The composition and nutrient levels of the diet are shown in Appendix A. Each pig in the GHRP group was s.c. injected with 3 mg/kg [D-Lys3]-GHRP-6 (synthesized by Hangzhou Taijia Biotech Co., Ltd., Hangzhou, China) at 8:00 and 13:00 each day, whereas pigs in the CON group were s.c. injected with the same volume of saline (0.9% NaCl, pH 7.4) at the same time. Cameras were installed on the ceiling above the metabolic cage to observe the intake behavior of pigs during the experiment.

### 2.2. Sample Collection

The initial and final body weights and feed intakes during the experiment were recorded to determine the average daily gain (ADG), average daily feed intake (ADFI) and the ratio of feed to gain (F/G). On day 6, jugular blood (total 10 mL; 2 mL was collected in tubes containing EDTA and a protease inhibitor to prevent the degradation of the ghrelin) was collected every 4 hours from 2:00 a.m. to 22:00 p.m. (2 people controlled the pig’s limbs and head, and then another person who was very familiar with blood collection used vacuum blood collection vessels to collect blood samples) and was immediately centrifuged at 3000 rpm for 10 min, and the supernatant was stored at −80 °C for the determination of ghrelin and amino acid (6:00 and 18:00) concentrations in the supernatant within 2 days. On day 8, all pigs were euthanized at 8:00 a.m. The proximal colonic digesta were collected and stored at −80 °C for the determination of short-chain fatty acids and the analysis of microbial composition. The tissue sections of longissimus dorsi at the 8th thoracic vertebrae and the left gastrocnemius were immersed in 4% paraformaldehyde for morphological analysis. In addition, tissue samples of longissimus dorsi and gastrocnemius were collected and stored at −80 °C for gene expression analysis.

### 2.3. Analysis of Feeding Behavior

A video of the feeding behavior of the pigs from 7:00 to 18:00 during the experiment was collected through the camera, and then the feeding times of the pigs were observed and calculated every two hours. Feeding behavior was defined as the pig’s head being above or within the feeder for at least 5 s.

### 2.4. Morphometric Analysis

The tissue samples of longissimus dorsi and gastrocnemius used for morphological observations were taken out and embedded with paraffin according to the conventional method and were then sectioned (perpendicular to muscle fibers) at 4 μm [22]. A total of 5–6 photomicrographs were taken for each slide, and 30–40 fibers were measured for each photomicrograph. Cross sectional areas (CSA) of longissimus dorsi and gastrocnemius fibers were measured using Image-Pro Plus 6.0 software from the photomicrographs.

### 2.5. Concentrations of Ghrelin and Amino Acids in Serum

The concentration of ghrelin in serum was determined according to the kit instructions (Jiancheng Bioengineering Institution, Nanjing, China). The determination of serum amino acid concentration was completed by Suzhou panomic Biomedical Technology Co., Ltd. (Suzhou, China). Briefly, 200 μL serum were transferred into a 2 ml EP tube, and then fully mixed with 400 μL 10% formic acid methanol-H_2_O (1:1. *v*/*v*) solution, followed by centrifugation at 12,000 rpm at 4 °C for 5 min. Next, after mixing 10 μL centrifuged supernatant with 490 μL 10% formic acid methanol-H_2_O (1:1. *v*/*v*) solution, we took 100 μL diluted samples and added 100 μL internal standard (13 C-labelled Phe, 100 ppb) into them. Finally, the mixed solution were filtered with the 0.22μM-pore-size membrane for subsequent detection by the liquid chromatography-tandem mass spectrometric (LC-MS) method. The LC-MS instrument consisted of a Waters ACQUITY UPLC system (Waters, Milford, Massachusetts, USA), equipped with an analytical C18 column (2.1 × 100 mm, 1.7 μm, Waters, Milford, MA, USA) and an AB 4000 triple quadrupole mass spectrometer.

### 2.6. Short-Chain Fatty Acid (SCFA) Concentration

The SCFA (acetate, propionate, isobutyrate, butyrate, isovalerate and valerate) concentrations in the proximal colon were determined by a capillary column gas chromatograph (GC-14B, Shimadzu, Japan; Capillary Column: 30 m × 0.32 mm × 0.25 µm film thickness), according to the previous method [23].

### 2.7. DNA Extraction, MiSeq Sequencing and Data Processing

The total genomic DNA from proximal colonic digesta samples was extracted by the phenol–chloroform method, according to the previous method [24]. The sequencing was finished by Shanghai Biozeron Biotechnology Co., Ltd. (Shanghai, China). The V3-V4 region of the bacteria 16S rRNA gene were amplified by PCR using primers 341F 5′-CCTAYGGGRBGCASCAG-3′ and 806R 5′-GGACTACNNGGGTATCTAAT 3′, according to previous methods [25]. The amplified products were purified using the AxyPrep DNA Gel Extraction Kit (Axygen Biosciences, Union City, CA, USA) according to the manufacturer’s instructions and were then pooled in equimolar and paired-end sequenced (2 × 250) on an Illumina MiSeq platform according to the standard protocols. Operational taxonomic units (OTUs) were clustered with a 97% similarity cutoff using UPARSE (version 10.0; http://drive5.com/uparse/, CA, USA, accessed on 1 August, 2021), and chimeric sequences were identified and removed using UCHIME. The phylogenetic affiliation of each 16S rRNA gene sequence was analyzed by the uclust algorithm (http://www.drive5.com/usearch/manual/uclust_algo.html, accessed on 1 August 2021) against the silva (SSU138.1) 16S rRNA database using a confidence threshold of 80% [26]. The diversity indices, including Chao1, Simpson and Shannon diversity indices, were assessed using the MOTHUR (version 1.36.1, MI, USA) [27]. Principal coordinate analysis (PCoA) was conducted based on the Bray–Curtis distance [28]. Linear discriminant analysis effect size (LEfSe) analysis was used for the quantitative analysis of biomarkers for highly dimensional colonic bacteria [29]. To identify the differential OTUs, we combined two methods of LEfSe (LDA > 2, *p* < 0.05) and Deseq2 [30] (|Log2 Fold Change| > 2, *p* < 0.05). OTUs of significant difference in both of the two methods were used for further analysis. These sequencing datasets are available from NCBI project SPR365829.

### 2.8. RNA Extraction, cDNA Synthesis and Quantitative Real-Time PCR

Total RNA was extracted from the longissimus dorsi and gastrocnemius samples of pigs with RNAiso Plus Total RNA extraction reagent (Takara Bio, Shiga, Japan), according to the instructions. An amount of 2 μL of total RNA was taken, and primescript^®^ RT Kit (Takara bio, Shiga, Japan) was used to synthesize complementary DNA (cDNA). PCR reactions were performed using the Roche SYBR Green PCR kit (Roche Hercules, CA, USA) (the PCR reaction mixture volume was 20 μL), according to the manufacturer’s instructions. The cDNA of muscle samples was analyzed by the ABI 7300 real-time PCR system (SDS, foster, CA, USA). Primer sequences are listed in Appendix A. The relative RNA expression of the target genes was calculated by formula 2^−ΔΔCt^ [31].

### 2.9. Statistical Analysis

The data were analyzed by SPSS version 24 software (SPSS Inc., Chicago, IL, USA) as a completely randomized design, considering the GHRP treatment as the main effect and the replicate as a block. A pen was considered as an experimental unit (i.e., n = 6). The differences of growth performance, muscle fiber CSA, ghrelin, amino acids, feeding behavior, SCFAs and mRNA between the groups were evaluated by independent sample *t*-tests. The alpha diversity indices of bacterial communities were compared by the Wilcoxon rank-sum test. Taking *p* < 0.05 as the standard, it showed that there were significant differences. Data visualization was performed by using GraphPad Prism (version 8.0.1, GraphPad Software, San Diego, CA, USA).

## 3. Results

### 3.1. The Growth Performance and Muscle Morphological Changes

Compared with the control group, pigs injected with [D-Lys3]-GHRP-6 showed a significant decrease in average daily gain (*p* < 0.05) (Figure 1A) and increase in the F/G (*p* < 0.05) (Figure 1B). However, there were no differences in feed intake between the two groups (Figure 1C). The injection of [D-Lys3]-GHRP-6 significantly reduced the cross-sectional area of the longest dorsal muscle (*p* < 0.05) (Figure 1D) and gastrocnemius muscle (*p* < 0.05) (Figure 1E) fibers in pigs.

### 3.2. Concentrations of Ghrelin and Amino Acids in Serum

The serum ghrelin concentrations at 6:00 and 22:00 in pigs of the GHRP group, measured on the sixth day of injection, were lower than that in the CON group (*p* < 0.05) (Figure 2). The levels of serum isoleucine, methionine and tyrosine at 6:00 were significantly decreased in the GHRP group than in the CON group (all *p* < 0.05). In addition, the level of the serum tryptophan tended to decrease in the GHRP group compared with that of the CON group (*p* = 0.083) (Figure 3A). The levels of serum arginine and tyrosine at 18:00 were significantly decreased in the GHRP group compared to that of the CON group (all *p* < 0.01), and the serum alanine level was significantly increased compared to the CON group (*p* < 0.05) (Figure 3B).

### 3.3. Feeding Behavior

The total feeding time of the GHRP group from 8:00 to 11:00 was lower than that of the CON group (*p* < 0.05) (Figure 4A). The statistical results of the feeding time every 2 hours during feeding show that the feeding time of the GHRP group from 9:00 to 11:00 after injection was significantly lower than that of the CON group (*p* < 0.05) (Figure 4B). When comparing the feeding frequency, it was also found that the total feeding frequency of the GHRP group between 8:00 and 11:00 tended to reduce compared with that of the CON group (*p* = 0.067) (Figure 4C). Similarly, the feeding frequency of the GHRP group from 9:00 to 11:00 after injection also was lower than that of CON group (*p* < 0.05) (Figure 4D).

### 3.4. SCFA Concentrations and Microbial Composition

As shown in Table 1, the concentrations (*p* < 0.05) and molar ratios (*p* < 0.05) of acetate in the colonic digesta of pigs were significantly reduced by the injection of GHRP. However, the concentrations (*p* < 0.01) and molar ratios (*p* < 0.05) of valerate in the colonic digesta of pigs in the GHRP group were significantly higher than in the CON group. There were no differences in the concentrations and molar ratio of other SCFAs between the two groups.

The 16S rRNA gene MiSeq sequencing shows that the rarefaction curves generated by MOTHUR plotting the number of OTUs by the number of sequences tended to approach the saturation plateau (Appendix A). Then, the α-diversity, as depicted by the Chao 1, Shannon and Simpson diversity indices, was analyzed (Figure 5A). The Chao 1 analysis, which predicts the number of taxa in a sample, shows a significant variation of microbiota richness between the colonic microbial composition of the GHRP and CON groups. Diversity indices, including the Shannon and Simpson indices, reveal no significant differences between the two groups. Principal coordinate analysis (PCoA) of unweighted UniFrac distances shows that there was a significant separation between the communities of the GHRP and CON groups (Figure 5B).

At the phylum level, the most abundant phylum detected in the proximal colonic digesta were Firmicutes, followed by Bacteroidetes and Actinobacteria. The injection of [D-Lys3]-GHRP-6 had no effect on the relative abundances of phyla (Appendix A). Linear discriminant analysis (LDA) effect size (LEfSe) analysis (LDA = 3.5) of colonic microbiota at the genus level shows that Prevotella_9, Limosilactobacillus, Anaerovibrio, Megasphaera, etc., a total of 9 genera, were more abundant in the GHRP group than in the CON group, whereas Oscillospiraceae UCG-005, Parabacteroides, unclassified Clostridia UCG-014, Oscillospiraceae NK4A214 group, etc., a total of 13 genera, were more abundant in the CON group (Figure 5C).

At the OTU level, we combined two methods of LEfSe and Deseq2 to select a total of 157 differential OTUs (Supplementary File). According to their relative abundance, the top 10 OTUs significantly enriched in the CON or GHRP groups were analyzed (Table 2). In agreement with the results at the genus level, the injection of [D-Lys3]-GHRP-6 significantly decreased the relative abundances of OTUs annotated to *Parabacteroides*, *Oscillospiraceae* UCG-005 and *Oscillospiraceae* NK4A214 group, and it increased the relative abundances of OTUs annotated to *Limosilactobacillus*, *Prevotella_9* and Anaerovibrio.

### 3.5. Expression of Protein-Deposition-Related Genes

The mRNA expression of mTOR (*p* < 0.05) in longissimus dorsi muscles in the GHRP group was significantly down-regulated compared with the CON group, whereas no significant difference in the mRNA expression of mTOR, S6K1 and rpS6 in gastrocnemius was found between the two groups (Figure 6A). In addition, the mRNA expression of MCT-1 (*p* < 0.05) of the GHRP group was significantly down-regulated, and the mRNA expression of HDAC1 (*p* = 0.080) tended to up-regulate. PPAR-δ (*p* = 0.097) tended to down-regulate compared with the CON group in the longissimus dorsi (Figure 6B). However, in gastrocnemius muscles, compared with the CON group, only the mRNA expression of HDAC1 (*p* < 0.05) was significantly up-regulated in the GHRP group. The mRNA expression of key autophagy markers FoxO1 and *Beclin-1* in longissimus dorsi muscles of the GHRP group were significantly up-regulated (all *p* < 0.05), whereas there was no significant difference between the two groups in gastrocnemius muscles. Then, the mRNA expression of differentiation, the myogenic genes MyoD and MYOG, and the muscle atrophy and degradation genes Atrogin-1 and MuRF-1 in longissimus dorsi and gastrocnemius were measured (Figure 6C). In longissimus dorsi muscles, the mRNA expression of MyoD (*p* < 0.05) in the GHRP group significantly down-regulated, and Atrogin-1 (*p* < 0.05) significantly up-regulated compared with the CON group. In addition, in gastrocnemius muscles, the mRNA expression of Atrogin-1 (*p* < 0.05) in the GHRP group was also significantly up-regulated compared with the CON group.

## 4. Discussion

Muscle homeostasis is regulated by protein synthesis processes such as muscle proliferation, differentiation and fusion, as well as protein catabolism [32]. In the last decade, studies have found that ghrelin plays an important role in promoting muscle development and alleviating muscle deficiency [5,33,34]. [D-Lys3]-GHRP-6, as a growth hormone secretagogue receptor (GHSR) antagonist, is often used to study the traits affected by ghrelin [35,36,37]. In this study, a pig model of inhibiting the effect of ghrelin was successfully established by injecting ghrelin antagonist ([D-Lys3]-GHRP-6). The eating habits of pigs are different from those of people, that pigs generally do not eat all the feed at once. Our previous research found that, after feeding at 7:00 a.m., they presented feeding behaviors until 5:00 p.m. In addition, there are no regular changes in pig feeding behavior and serum ghrelin levels [38]. People’s normal eating times are 7:00, 12:00 and 17:00. Therefore, according to people’s normal eating times, we chose to inject at 8:00 and 13:00. However, we wanted the antagonist to more evenly affect the feeding behavior of pigs throughout the day, and 7:00 a.m. is the starting feeding time of pigs. In order to prevent conflict with feeding, we chose to start injection at 8:00 a.m. A recent study found that ghrelin treatment improves muscle atrophy more significantly under fasting than under feeding [3]. In order to more accurately explore the mechanism of ghrelin on protein deposition in muscles, we adopted the method of feeding restriction (90% of ad libitum) for experimental pigs. Therefore, no significant difference in average daily food intake was found between the two groups in this study. However, we found that the injection of [D-Lys3]-GHRP-6 significantly reduced the growth performance of pigs, indicating that ghrelin plays an important role in the growth and development of pigs, which is consistent with the previous research results of chickens [39] and fish [40]. A previous study showed that the muscle fiber cross-sectional area (CSA) distribution of anterior tibial muscle in TG mice shifted slightly to a larger area compared with WT mice, indicating that unacylated ghrelin has a protective effect on muscle loss [41]. Here, we also report a significant reduction in muscle fiber cross-sectional area (CSA) of longissimus dorsi and gastrocnemius in the GHRP group compared with the CON group.

Feeding behavior includes homeostatic feeding and hedonic feeding. Hedonic feeding is mainly stimulated by vision or smell, whereas homeostatic feeding is controlled by circulating hormones in the hypothalamus [42]. Ghrelin treatment can increase the food intake of animals [6,43]. Human plasma ghrelin levels increase before meals and decrease after meals, suggesting that ghrelin can regulate animal feeding behavior as a hunger signal [8]. Therefore, we wanted to explore the effect of ghrelin on the feeding behavior of pigs. Under the same daily feeding amount, the effect of the injection of [D-Lys3]-GHRP-6 on the feeding behavior of pigs supports that the antagonization of ghrelin has the function of inhibiting feeding.

A surge of amino acids activates the translation initiation factor of pigs and stimulates the signal of protein synthesis [44,45]. In addition, changes in serum amino acids in pigs fed continuously and intermittently show that changes in feeding behavior affect amino acid metabolism [10,43]. In this study, the injection of [D-Lys3]-GHRP-6 significantly changed the serum amino acids levels at 6:00 and 18:00. The interaction between arginine, isoleucine and their sensors promote the binding of Rag protein and mTOR and then promote the synthesis of protein [46,47]. Likewise, methionine can promote mTORC1-dependent protein synthesis through its metabolite, S-adenosylmethionine (SAM) [48]. The increased alanine concentration may be due to the decomposition of muscle protein, resulting in the increased bioavailability of amino acids in response to the stress caused by the lack of gut microbiota [17]. In our study, we did find that the gene expression of *mTOR* significantly down-regulated after the injection of [D-Lys3]-GHRP-6, which indicates that the antagonization of ghrelin can block the pulsatile pattern of amino acids, leading to the inhibition of the mTOR signal pathway, finally reducing the synthesis of proteins in muscles.

Many studies have shown that the interaction between host and gut microbiota is strictly regulated to maintain body homeostasis [49,50]. Recently, a study found for the first time that ghrelin can regulate the composition of gut microbiota, and gut microbiota of the Ghrl^−/−^ mice develops in a direction that is not conducive to muscle development [3]. Here, we also found that the injection of [D-Lys3]-GHRP-6 significantly affects the colonic microbial diversity and composition of pigs. This study shows a significant reduction in the abundance of acetate-producing bacteria, *Oscillospiraceae* UCG-005, *Parabacteroides* and *Oscillospiraceae* NK4A214 group in the GHRP group compared to the CON group. Many interactions between host and gut microbiota are mediated by SCFAs produced by bacterial fermentation of dietary polysaccharides [51]. Here, we show that the injection of [D-Lys3]-GHRP-6 significantly reduced the concentration and molar ratio of acetate in the colonic digesta of pigs. The SCFAs produced by the distal colon can avoid liver metabolism and directly enter systemic circulation to act on various tissues [52]. SCFAs such as acetate, propionate and butyrate have been recognized as HDAC (which regulate gene transcription by removing the acetyl groups on histone lysine residues) inhibitors in various tissues [18]. It was found that, in the case of nutritional deficiency, HDAC1 activates autophagy, thereby promoting skeletal muscle atrophy [20]. The results of gene quantification in this study show that the mRNA expression of *MCT-1* (monocarboxylate transporter, and the key element to transport SCFAs in various tissues) were significantly down-regulated, whereas that of *HDAC1*, *FOXO1* and *Beclin-1* were significantly up-regulated in the GHRP group compared with the CON group, indicating that the antagonization of ghrelin can indeed reduce the concentration of acetate, an HDAC1 inhibitor, thereby promoting autophagy and ultimately reducing protein deposition.

Protein deposition in muscles is directly regulated by myogenic regulatory factors and muscle degradation factors. MyoD and MyoG are members of the myogenic regulatory factor protein family, which can promote muscle proliferation and differentiation [53]. Atrogin-1 and MuRF-1 are ubiquitin ligases expressed in muscles, which can induce polyubiquitination of proteins and promote protein hydrolysis [54]. Here, we show that the injection of [D-Lys3]-GHRP-6 significantly down-regulated the mRNA expression of *MyoD* in muscles and up-regulated the gene expression of *Atrogin-1*, which further illustrates the negative role of the antagonization of ghrelin in protein deposition in muscles.

## 5. Conclusions

In conclusion, we inversely demonstrated the important role of ghrelin in maintaining protein content in muscles by injecting [D-Lys3]-GHRP-6 in a short term. Our study proves that the antagonization of ghrelin may affect the eating patterns of pigs, which altered the colonic microbiota and their metabolites (SCFAs), leading to the activation of autophagy, thereby reducing protein deposition in muscles. In addition, the antagonization of ghrelin also affected the absorption of amino acids, which resulted in inhibiting the mTOR pathway and reducing protein synthesis in muscles. This study suggests that the antagonization of ghrelin is not conducive to protein deposition in muscles, suggesting that ghrelin may have broad prospects in preventing muscle loss and promoting muscle development.

## Figures and Tables

**Figure 1 biology-11-00840-f001:**
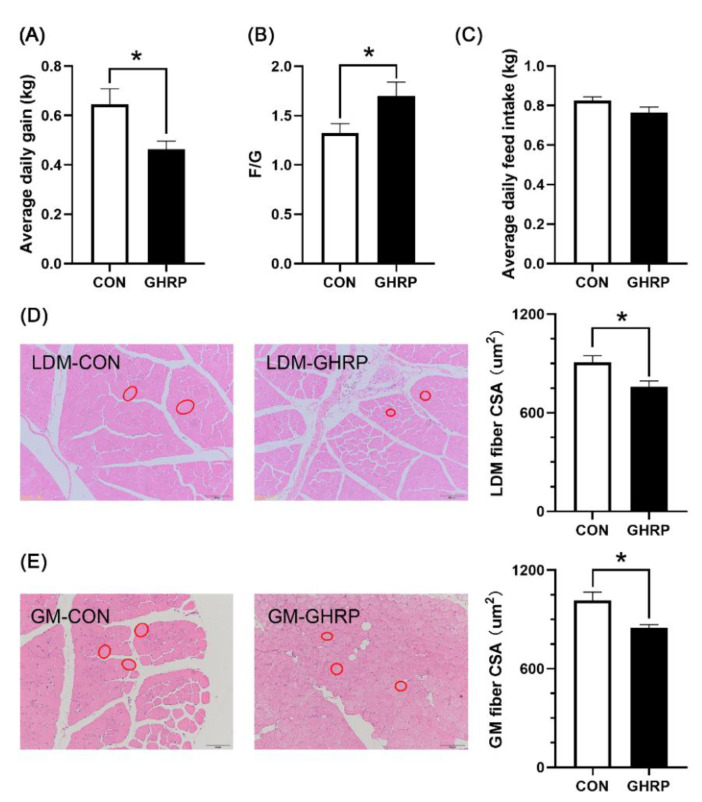
Effects of injection of ghrelin antagonist ([D-Lys3]-GHRP-6) on the growth performance and morphology of muscles in pigs. Average daily gain (**A**), feed conversion ratio (**B**) and average daily feed intake (**C**) over the 7 d trial period, and the cross-sectional areas (CSA) of longissimus dorsi muscle (**D**) and gastrocnemius muscle (**E**) fibers were measured after 7 d of injection of [D-Lys3]-GHRP-6 (GHRP) or saline (CON). Values are means ± SEMs; n = 6. * *p* < 0.05 versus control group. LDM: longissimus dorsi muscle; GM: gastrocnemius muscle. The red circle indicates the cross section of muscle fibers.

**Figure 2 biology-11-00840-f002:**
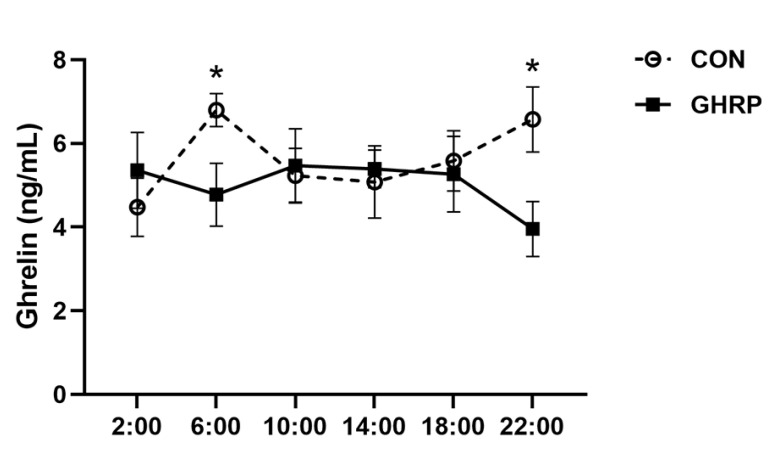
Changes in the level of ghrelin in serum after injection of ghrelin antagonist ([D-Lys3]-GHRP-6). The injections were performed at 8:00 and 13:00. Values are means ± SEMs; n = 6. * *p* < 0.05 versus control group. CON: injection of saline; GHRP: injection of [D-Lys3]-GHRP-6.

**Figure 3 biology-11-00840-f003:**
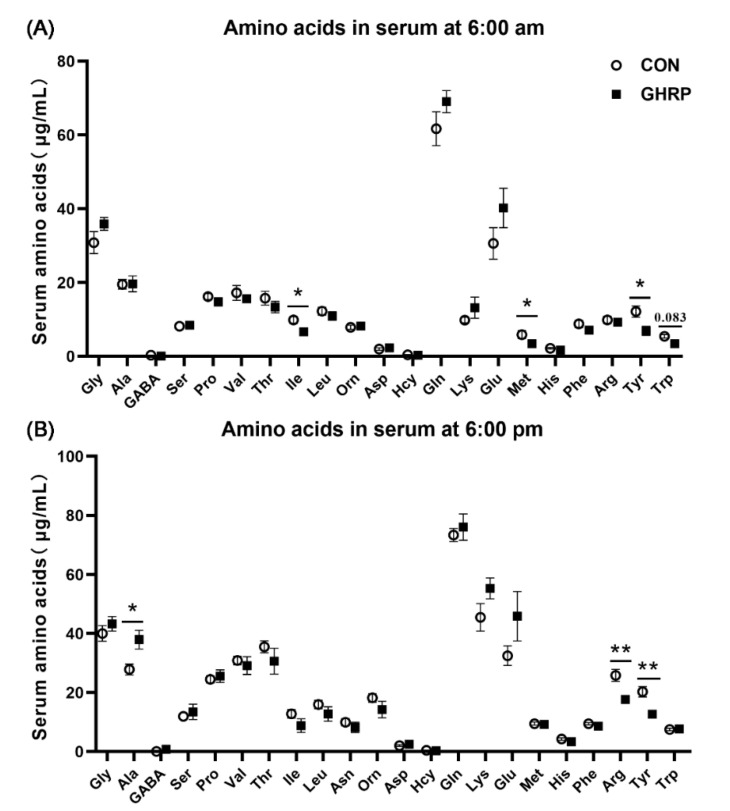
The serum concentrations of amino acids of pigs in the control group (CON) and ghrelin antagonist group (GHRP) at 6:00 (**A**) and 18:00 (**B**). Values are means ± SEMs; n = 6. * *p* < 0.05 versus control group; ** *p* < 0.01 versus control group.

**Figure 4 biology-11-00840-f004:**
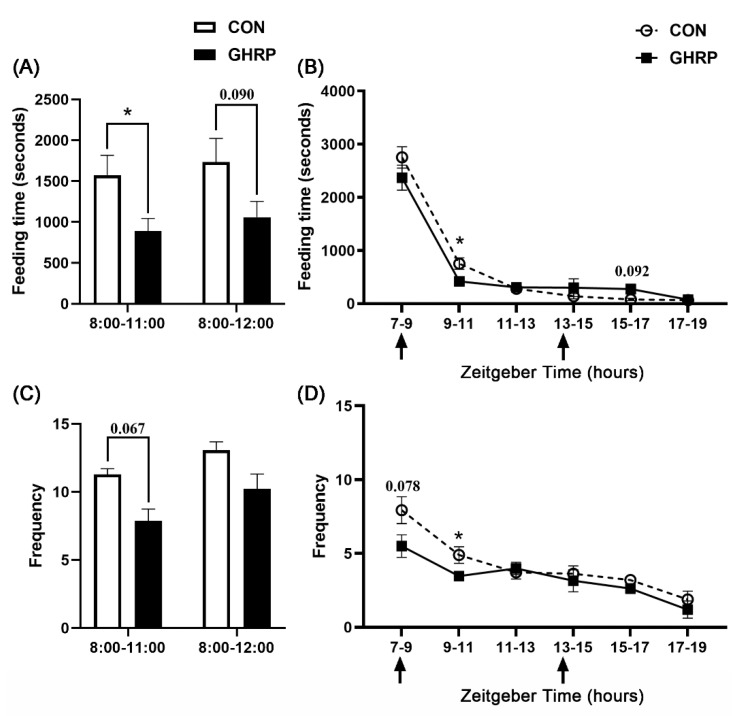
Injection of ghrelin antagonist ([D-Lys3]-GHRP-6) affected the feeding behavior of pigs. The total feeding time (**A**) and frequency (**C**) from 8:00 to 11:00 and from 8:00 to 12:00, and the feeding time (**B**) and frequency (**D**) every 2 hours from 7:00 to 19:00 over the 7 d trial period, were counted. Values are means ± SEMs; n = 6. * *p* < 0.05 versus control group. The arrow indicates injection of [D-Lys3]-GHRP-6 (GHRP) or saline (CON).

**Figure 5 biology-11-00840-f005:**
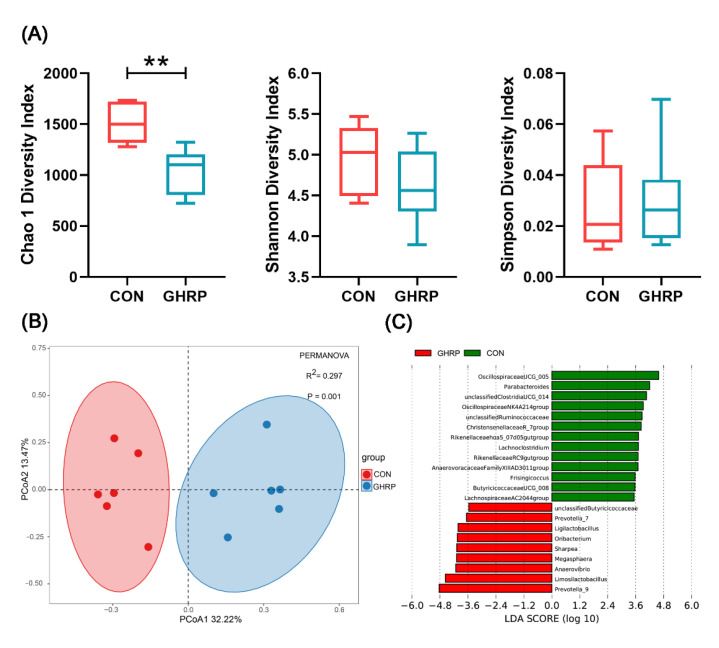
The injection of ghrelin antagonist ([D-Lys3]-GHRP-6) altered the colonic microbiota of pigs. (**A**) Comparison of alpha diversity metrics for the colonic microbiome between the CON and GHRP groups. Values are means ± SEMs; n = 6. ** *p* < 0.01 versus the CON group. (**B**) Beta diversity comparison for the colonic microbiome between the CON and GHRP groups. Unweighted UniFrac principal coordinate analysis plot of colonic bacteria based on the Bray–Curtis similarity clustering analysis of the abundance of OTUs. Axes represent two synthetic variables explaining the greatest proportion of variations in the samples. (**C**) Linear discriminant analysis (LDA) effect size (lefse) analysis results of colonic microbiota at the genus level. LDA (linear discriminant analysis) plot indicates biomarkers found by ranking accordingly to their effect size (3.5) of the genus.

**Figure 6 biology-11-00840-f006:**
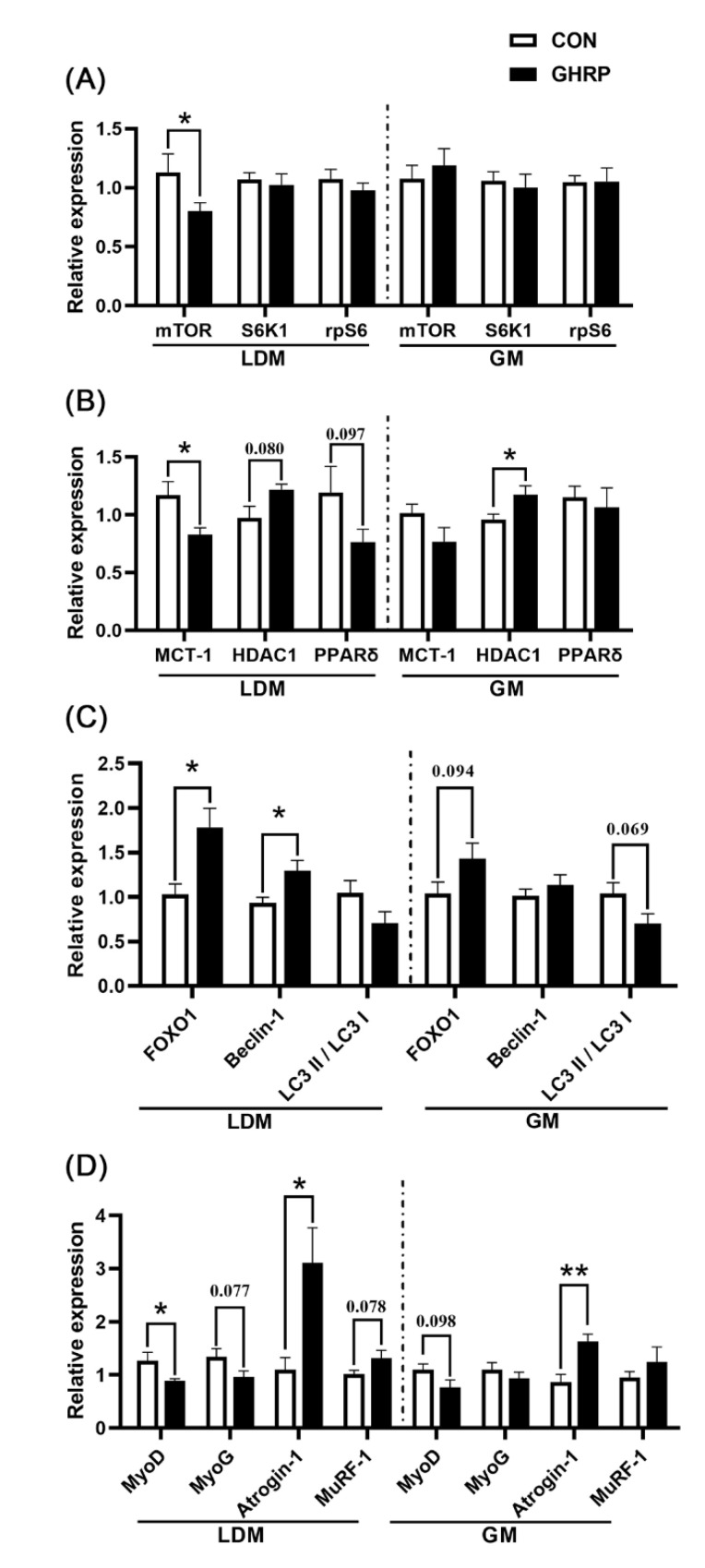
The mRNA expression of genes related to protein deposition in the control group (CON) and ghrelin antagonist group (GHRP). (**A**) Relative expression levels of genes related to protein synthesis in mTOR pathway (mTOR, S6K1, rpS6) in longissimus dorsi muscles and gastrocnemius muscles. (**B**) The expression of genes MCT-1, HDAC1 and PPAR-δ in longissimus dorsi muscles and gastrocnemius muscles. (**C**) The expression of genes FOXO1 and *Beclin-1* and LC3 II to LC3 I ratio in longissimus dorsi muscle and gastrocnemius muscle. (**D**) The expression of genes MyoD, MyoG, Atroigin-1 and MuRF-1 in longissimus dorsi muscles and gastrocnemius muscles. Values are means ± SEMs; n = 6. * *p* < 0.05 versus control group; ** *p* < 0.01 versus control group. LDM: longissimus dorsi muscles; GM: gastrocnemius muscles.

**Table 1 biology-11-00840-t001:** Effect of injection of ghrelin antagonist ([D-Lys3]-GHRP-6) on the concentration and proportion of SCFAs in colonic digesta of pigs.

Item	CON	GHRP	*p*-Value
Concentration (mmol/g)			
Acetate	33.58 ± 2.48	26.93 ± 1.61	0.048
Propionate	13.79 ± 1.59	15.47 ± 1.51	0.462
Iso-butyrate	0.96 ± 0.08	1.29 ± 0.20	0.158
Butyrate	5.52 ± 0.67	5.56 ± 0.45	0.965
Iso-valerate	1.70 ± 0.15	2.13 ± 0.34	0.275
Valerate	1.06 ± 0.05	2.34 ± 0.42	0.029
Total SCFAs	56.62 ± 4.49	53.73 ± 2.49	0.585
Molar proportions of SCFAs (mol%)			
Acetate	59.44 ± 0.84	50.23 ± 2.21	0.003
Propionate	24.04 ± 1.01	28.57 ± 1.91	0.072
Iso-butyrate	1.80 ± 0.27	2.46 ± 0.39	0.195
Butyrate	9.65 ± 0.58	10.32 ± 0.64	0.454
Iso-valerate	3.14 ± 0.43	4.08 ± 0.68	0.270
Valerate	1.93 ± 0.17	4.34 ± 0.74	0.022

Values are means ± SEMs; n = 6. CON, injection of saline; GHRP, injection of [D-Lys3]-GHRP-6.

**Table 2 biology-11-00840-t002:** Effect of injection of ghrelin antagonist ([D-Lys3]-GHRP-6) on the bacterial OTUs in colonic digesta of pigs.

OTU ID	LDA−score	Deseq−FC	CON	GHRP	Annotation
Significantly enriched in CON group					
OTU28	−3.35	−3.32	1.04 ± 0.44	0.22 ± 0.18	[Ruminococcus] gauvreauii group
OTU24	−3.36	−5.28	1.00 ± 0.48	0.06 ± 0.04	*Oscillospiraceae* UCG−005
OTU21	−3.58	−8.47	1.49 ± 0.50	0.01 ± 0.01	uncultured Muribaculaceae
OTU63	−3.25	−5.01	0.72 ± 0.22	0.04 ± 0.03	uncultured Ruminococcaceae
OTU9	−3.90	−7.72	3.87 ± 2.32	0.05 ± 0.04	*Oscillospiraceae* UCG−005
OTU7	−3.81	−3.42	2.96 ± 0.98	0.50 ± 0.39	*Parabacteroides*
OTU2	−3.99	−4.93	5.14 ± 3.27	0.44 ± 0.40	*Prevotella*
OTU51	−3.22	−4.92	0.74 ± 0.22	0.06 ± 0.04	*Oscillospiraceae* NK4A214 group
OTU52	−3.27	−4.47	0.79 ± 0.33	0.05 ± 0.02	*Lachnoclostridium*
OTU57	−3.34	−4.91	1.07 ± 0.39	0.07 ± 0.06	Rikenellaceae gut group
Significantly enriched in GHRP group					
OTU25	3.84	10.45	0.00 ± 0.00	2.29 ± 1.32	*Megasphaera*
OTU27	3.52	4.50	0.07 ± 0.03	1.52 ± 0.63	*Prevotella_9*
OTU26	3.49	8.67	0.00 ± 0.00	1.66 ± 0.60	*Sharpea*
OTU20	3.61	5.92	0.02 ± 0.00	2.02 ± 0.64	*Ligilactobacillus*
OTU23	3.63	4.10	0.07 ± 0.04	1.74 ± 1.07	*Anaerovibrio*
OTU6	4.09	4.53	0.20 ± 0.08	5.17 ± 2.23	*Prevotella_9*
OTU4	4.04	4.69	0.21 ± 0.07	5.35 ± 2.60	*Prevotella_9*
OTU58	3.64	9.97	0.00 ± 0.00	1.44 ± 0.79	*Olsenella*
OTU54	3.63	9.96	0.00 ± 0.00	1.50 ± 1.03	[Ruminococcus] gauvreauii group
OTU34	3.47	3.02	0.09 ± 0.02	1.21 ± 0.58	*Limosilactobacillus*

Values are means ± SEMs; n = 6. CON, injection of saline; GHRP, injection of [D-Lys3]-GHRP-6.

## Data Availability

These sequencing datasets are available from NCBI project SPR365829.

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
