# Peer review of "Antagonization of Ghrelin Suppresses Muscle Protein Deposition by Altering Gut Microbiota and Serum Amino Acid Composition in a Pig Model"

_biology, 2022, doi:10.3390/biology11060840_

Round 1

Reviewer 1 Report

This study investigated the effects of the antagonization of ghrelin on the eating pattern, gut microbiota and muscle protein deposition in a pig model by injecting ghrelin antagonists in a short term. The authors found that antagonization of ghrelin suppressed muscle protein deposition by altering gut microbiota and serum AA composition Generally, this is a novel study with well-writing. However, there are a number of issues the authors should address.
Ghrelin is an appetite stimulating hormone that can increase food intake, however, in this study, antagonization of ghrelin had no effect on the food intake of pigs. Please clarify in the discussion section.
In this study, pigs were fed 90% of ad libitum. Please clarify why, and pay attention to this point in this manuscript. In addition, there is no information on the diet used in this study.
The last part of the introduction needs to clarify the innovation of this research.
The names of bacteria at genus level and gene abbreviations should be in italics.
"Regulate" in the manuscript should be more specific, such as increased or decreased.
The authors compared the bacterial community between two groups at the phylum and genus levels, do you analyze the data at the OTU level?
Attention should be paid to the logic and coherence of the discussion part in the manuscript.

Author Response

  1. Ghrelin is an appetite stimulating hormone that can increase food intake, however, in this study, antagonization of ghrelin had no effect on the food intake of pigs. Please clarify in the discussion section.

Response: In our study, the feeding restriction was adopted for the experimental pigs, so there was no significant difference in the average daily feed intake between the two groups.

  1. In this study, pigs were fed 90% of ad libitum. Please clarify why, and pay attention to this point in this manuscript. In addition, there is no information on the diet used in this study.

Response: A recent study found that ghrelin treatment improved muscle atrophy more significantly under fasting than under feeding. Therefore, in order to more accurately explore the mechanism of ghrelin on protein deposition in muscle, we adopted the method of feeding restriction (90% of ad libitum) for experimental pigs.

  1. The last part of the introduction needs to clarify the innovation of this research.

Response: I have added the innovation of this paper in the last part of the introduction.

  1. The names of bacteria at genus level and gene abbreviations should be in italics.

Response: I have modified the format of these genera and genes

  1. "Regulate" in the manuscript should be more specific, such as increased or decreased.

Response: I have made appropriate changes to the "regulate" used in this article.

  1. The authors compared the bacterial community between two groups at the phylum and genus levels, do you analyze the data at the OTU level?

Response: I also analyzed the bacterial community at OTU level, and the analysis results have been put in the manuscript.

  1. Attention should be paid to the logic and coherence of the discussion part in the manuscript.

Response: Thanks for the valuable comments, we revised the discussion part in the new version.

Reviewer 2 Report

1. Keep consistency in the use of abbreviations, e.g F: G and F/G. And check for misspellings in the text. 

2. How many fibers were measured for morphology analysis and how were they measured?

3. Ethical permit number should be written in the "Animals and Experimental Design section". What was the gender of the pigs?

4. Why were the timepoints 8:00 and 13:00 chosen for injections? Why not spread out the injections more evenly?

5. Some references are missing in the introduction section.

Author Response

  1. Keep consistency in the use of abbreviations, e.g F: G and F/G. And check for misspellings in the text.

Response: I have changed all "F: G" in the text to "F / G".

  1. How many fibers were measured for morphology analysis and how were they measured?

Response: I take 5-6 photomicrographs for each slide, and measure 30-40 fibers for each photomicrograph. Cross sectional area (CSA) of the fibers were measured using Image-Pro Plus 6.0 software from photomicrograph. And I have added this part to the text.

  1. Ethical permit number should be written in the "Animals and Experimental Design section". What was the gender of the pigs?

Response: I have written the ethical license number in the "animal and experimental design section". Barrows were used in the experiment, which has been added to the text.

  1. Why were the timepoints 8:00 and 13:00 chosen for injections? Why not spread out the injections more evenly?

Response: The feeding habits of pigs are different from those of people. Pigs will not eat all of them immediately after feeding. Our previous research found that after feeding at 7:00 in the morning, they have feeding behavior until 5:00 in the afternoon. In addition, there are no regular changes in pig feeding behavior and serum ghrelin level. People's normal eating time is 7:00, 12:00 and 17:00 respectively. Therefore, according to people's normal eating time, we choose to inject at 8:00 and 13:00. On the other hand, we want the antagonist to more evenly affect the feeding behavior of pigs throughout the day, and seven o'clock is the starting feeding time of pigs. In order to prevent conflict with feeding, we choose to start injection at 8:00.

  1. Some references are missing in the introduction section.

Response: I have added the required references to the introduction.

Reviewer 3 Report

The manuscript entitled "Antagonization of Ghrelin suppresses muscle protein deposition by altering gut microbiota and serum amino acid composition in a pig model " was designed to investigate the effects of the antagonization of ghrelin on muscle protein deposition, eating pattern and gut microbiota in pigs by injecting of D-Lys3]-GHRP-6, ghrelin antagonist. Authors concluded that antagonization of ghrelin reduced the concentration of Arg, Met, Ile, acetate which eventually reduced the protein deposition in muscle. The manuscript is well written, coherent, organized, and clearly presented. However, the following changes and questions need to be considered and answered, respectively.

  1. What was the dietary composition of the feed that was fed to the experimental pigs?
  2. Ghrelin in the blood sample can easily undergo degradation. What was added to the blood samples to prevent Ghrelin in the blood from undergoing degradation during the blood collection?
  3. How much blood volume was collected at each sampling point?
  4. Did authors insert catheters in pigs for serial blood collection? If yes, describe the catheterization method. If no, describe in details how the pigs were restrained for blood collection.
  5. Line 193: Delete the phrase “Throughout …………………..experiment,”
  6. Label the horizontal axis of all graphs.
  7. Line 313: Replace the phrase “In recent ten years” with “In the last decade”

Finally, because the sample size is only 6 pigs per treatments, I would suggest that authors would moderate the conclusions of the study.

Author Response

  1. What was the dietary composition of the feed that was fed to the experimental pigs?

Response: I have added the dietary composition of the feed to the “Supplementary materials”.

  1. Ghrelin in the blood sample can easily undergo degradation. What was added to the blood samples to prevent Ghrelin in the blood from undergoing degradation during the blood collection?

Response: We select people who are very familiar with blood collection for blood collection to ensure shorter collection time, and 2 mL blood samples were collected in tubes containing EDTA and a protease inhibitor to prevent the degradation of the ghrelin, and complete the determination within two days.

  1. How much blood volume was collected at each sampling point?

Response: We collected 10 mL of blood samples at each time point.

  1. Did authors insert catheters in pigs for serial blood collection? If yes, describe the catheterization method. If no, describe in details how the pigs were restrained for blood collection.

Response: When we used the catheter for serial blood collection in the previous experiment, we found that the catheter has a great impact on pigs and the nursing of the catheter is also difficult. Therefore, we chose to find someone who is very familiar with blood collection to use the vacuum blood collection vessel for blood sample collection, so as to ensure that the blood collection can be completed in a very short time, so as to reduce the impact on pigs.

  1. Line193: Delete the phrase “Throughout ………experiment,”.

Response: I have deleted the phrase “Throughout ………experiment,”.

  1. Label the horizontal axis of all graphs.

Response: I have added the horizontal axis to the required graphs.

  1. Line 313: Replace the phrase “In recent ten years” with “In the last decade”

Response: I have completed the modification as required.

  1. Finally, because the sample size is only 6 pigs per treatments, I would suggest that authors would moderate the conclusions of the study.

Response: We agree with you very much and we will moderate the conclusions of the study.